# Copper-Induced Stimulation of Ectophosphatase Activity of *Candida albicans*

**DOI:** 10.3390/pathogens14070667

**Published:** 2025-07-08

**Authors:** Anita Leocadio Freitas-Mesquita, Fabiano Ferreira Esteves, José Roberto Meyer-Fernandes

**Affiliations:** 1Leopoldo de Meis Institute of Medical Biochemistry, Center of Health Science, Federal University of Rio de Janeiro, Rio de Janeiro 21941-590, RJ, Brazil; anitaleocadio2@gmail.com (A.L.F.-M.); fabferest@gmail.com (F.F.E.); 2National Institute of Science and Technology in Structural Biology and Bioimaging, Rio de Janeiro 21941-590, RJ, Brazil

**Keywords:** *Candida albicans*, ectophosphatase, copper stimulation, metal modulation, fungal adhesion

## Abstract

*Candida albicans* is an opportunistic fungal pathogen that can cause superficial and life-threatening infections, particularly in immunocompromised individuals. Its ability to adhere to host cells is critical for colonization and infection. In this context, investigating ectophosphatases is particularly relevant, as these enzymes have been associated with fungal adhesion to host cells. This study aimed to investigate the nature of copper-induced stimulation of ectophosphatase activity in *C. albicans*. Ectophosphatase activity was measured using *p*-nitrophenyl phosphate as substrate. Micromolar concentrations of CuCl_2_ markedly stimulated ectophosphatase activity, and its response to reducing agents and metal chelators suggested that this modulation does not involve redox reactions. The significant differences between the biochemical properties of basal (Cu^2+^-independent) and Cu^2+^-dependent ectophosphatase activities suggest the presence of at least two distinct ectophosphatases in *C. albicans*. Cu^2+^-independent ectophosphatase activity presented an acidic profile and was insensitive to Mg^2+^, whereas Cu^2+^-dependent ectophosphatase activity exhibited an alkaline profile and was also stimulated by Mg^2+^. Both activities were negatively modulated by classical phosphatase inhibitors, but Cu^2+^-dependent ectophosphatase had lower sensitivity compared to the basal activity. These findings highlight the role of copper as a modulator of *C. albicans* ectophosphatase activity and suggest potential implications for fungal adaptation during infection.

## 1. Introduction

*Candida albicans* was recently classified as a critical fungal pathogen in the World Health Organization (WHO)’s global priority list. This fungus is usually a harmless commensal but has the potential to become invasive, infectious, and even fatal by targeting numerous organ systems and acting as an opportunistic pathogen, especially in vulnerable, critically ill, and immunocompromised individuals [1,2,3]. *C. albicans* can cause two major types of infections in humans: superficial infections, such as oral or vaginal candidiasis, and life-threatening systemic infections [4]. Oral candidiasis has emerged as an early clinical clue for diagnosing acquired immunodeficiency syndrome (AIDS) and an indicator of its severity [5].

To establish infection, these opportunistic pathogens must evade the immune system, survive and replicate in the host environment, and spread to new tissues [6]. The expression of proteins involved in adhesion and invasion contributes significantly to the pathogenic potential of *C. albicans*. Adhesion is influenced by multiple factors, including the composition of cell wall proteins and the physical and chemical characteristics of the cell surface [7]. In this context, investigating ectophosphatases is particularly relevant, as these enzymes have been related to the adhesion of fungi to host cells [6,8,9,10,11,12].

Ectophosphatases are external surface membrane proteins widely distributed among various microorganisms, including numerous fungal species [13]. Several biochemical characterization studies have shown that ectophosphatase activities are typically modulated by divalent cations [13]. Notably, in *Candida parapsilosis*, a strong stimulation of ectophosphatase activity was observed in the presence of increasing millimolar concentrations of CuCl_2_ [10].

Copper is essential in biological systems due to its ability to transit between cuprous (Cu^+^) and cupric (Cu^2+^) oxidation, facilitating many important electron transfer reactions. Therefore, it acts as a prosthetic group for several enzymes involved in oxidation–reduction reactions, such as superoxide dismutase (SOD) [2,14]. On the other hand, copper can also promote the formation of potentially harmful reactive oxygen species via redox cycling between Cu^2+^/Cu^+^ states in the presence of dioxygen and physiological reducing agents [15]. To maintain copper homeostasis and prevent toxicity, *C. albicans* has developed complex regulatory mechanisms, including copper transporters and metal-binding proteins such as metallothioneins that sequester excess copper to avoid cellular damage [16].

In the present study, we found that micromolar concentrations of CuCl_2_ markedly stimulated the ectophosphatase activity of *C. albicans*. Considering the potential involvement of ectophosphatases in fungal infection and the importance of copper in various cellular processes, we aimed to investigate the nature of this stimulation. Additionally, we evaluated the differential biochemical responses between the basal and Cu^2+^-dependent ectophosphatase activities. Given the distinct biochemical profiles observed, we hypothesized that there are at least two distinct ectophosphatases in *C. albicans*.

## 2. Materials and Methods

### 2.1. Microorganisms and Growth Conditions

The experiments of this study were performed using the CAI-4 strain of *Candida albicans* (genotype Δ*ura3**::imm434/*Δ*ura3::imm434*), an auxotrophic mutant for uridine obtained from the wild-type strain SC5314 [17]. This strain was used because its URA3-genetic construction facilitates molecular manipulations, which may be useful for future studies. *C. albicans* cells were cultivated at room temperature, under constant agitation, in Modified Sabouraud Medium (SAB-M; 2% glucose, 1% peptone, 0.5% yeast extract) supplemented with 50 µg/mL uridine. After 48 h of growth, the yeasts were washed three times with PBS (150 mM NaCl, 10 mM NaH_2_PO_4_, and 10 mM Na_2_HPO_4_, pH 7.2) and collected by centrifugation (1500× *g*, 10 min). Cell growth was estimated by counting the yeasts in a Neubauer chamber.

### 2.2. Materials

All reagents were obtained from E. Merck (D-6100 Darmstadt, Germany) or Sigma-Aldrich Chemical Co. (St. Louis, MO, USA). Distilled water was deionized using a Milli-Q resin system (Millipore Corp., Bedford, MA, USA) and was used for the preparation of all solutions.

### 2.3. Ectophosphatase Activity Measurements

The ectophosphatase activity of *C. albicans* was determined using *p*-nitrophenyl phosphate (*p*-NPP) as a substrate, measuring the rate of *p*-nitrophenol (*p*-NP) production. Intact cells (1 × 10^8^) were incubated for 1 h at room temperature in 0.5 mL of a reaction mixture containing, unless otherwise specified, 116 mM NaCl, 4 mM KCl, 5.4 mM D-glucose, 20 mM HEPES buffer (pH 7.4), and 5.0 mM *p*-NPP. Reactions were initiated by adding living cells or the substrate and stopet by adding 1.0 mL of 1 M NaOH. Absorbance was measured spectrophotometrically at 425 nm, using *p*-NP as a standard [18]. Ectophosphatase activity was calculated by subtracting nonspecific *p*-NPP hydrolysis measured in the absence of cells. Cell viability was assessed before and after incubations using the Trypan blue method [19]. Viability was not affected by the experimental conditions used in this study.

### 2.4. Statistical Analysis

All experiments were performed in triplicate, and similar results were obtained from at least three independent cell suspensions. Data were analyzed statistically using one-way ANOVA followed by Tukey’s test, with GraphPad Prism software (GraphPad Software Inc., Version 8.0.1, San Diego, CA, USA). Results were considered statistically significant at *p* < 0.05, which was indicated by different lowercase letters above the bars.

## 3. Results

The ectophosphatase activity of *Candida albicans* increased in response to micromolar concentrations of CuCl_2_, reaching a maximum value (39.28 ± 1.32 nmols *p*-NP × 10^−8^ cells × h^−1^) at 200 µM CuCl_2_ (Figure 1). Therefore, all subsequent experiments were performed in the absence or presence of 200 µM CuCl_2_. The ectophosphatase activity measured after 30 min of pre-incubation with the metal was not significantly different from that without pre-incubation, indicating no dependence on pre-incubation.

Further assays were conducted to investigate the nature of this stimulation. To determine whether reducing agents could prevent copper-induced stimulation, ectophosphatase activity was measured in the presence of ascorbate (Asc), cysteine (Cys), β-mercaptoethanol (β-ME), and glutathione (GSH). Except for ascorbate, all other reducing agents tested prevented stimulation when added to the medium prior to the metal. In contrast, basal ectophosphatase activity was not affected by their presence (Figure 2A). To assess the reversibility of the copper-induced stimulation of ectophosphatase activity, we added the divalent ion chelator ethylenediaminetetraacetic acid (EDTA) or the copper-specific chelator bathocuproinedisulfonic acid disodium salt (BCS) to the reaction medium (Figure 2B). Although ectophosphatase activities remained significantly higher than in cells not exposed to copper, they were substantially lower than the ectophosphatase activity measured in the continuous presence of 200 µM CuCl_2_ due to the absence of chelators. It is possible that the chelators did not completely remove copper ions previously bound to the enzyme, thus accounting for this statistically significant difference. We also tested the effect of pre-incubating the cells with EDTA (Figure 2C), which resulted in a similar response to that observed with the reducing agents (Figure 2A), namely the reversal of the stimulation induced by CuCl_2_. It is important to note that basal ectophosphatase activity was not affected by chelators, whether they were added during the course of the reaction (Figure 2B) or pre-incubated with the cells (Figure 2C). These results suggest that copper stimulation does not involve the irreversible modification of *C. albicans* ectophosphatase.

To explore possible differences in biochemical properties, subsequent experiments compared the basal activity—referred to as Cu^2+^-independent ectophosphatase activity—with the Cu^2+^-dependent ectophosphatase activity of *C. albicans*. The Cu^2+^-dependent ectophosphatase activity was calculated by subtracting the Cu^2+^-independent activity from the activity measured in the presence of 200 µM CuCl_2_.

As shown in Figure 3, Cu^2+^-independent ectophosphatase activity increased under acidic pH conditions, whereas Cu^2+^-dependent ectophosphatase activity exhibited an alkaline profile. Unlike Cu^2+^-independent ectophosphatase activity, Cu^2+^-dependent ectophosphatase activity was stimulated by MgCl_2_, with maximal stimulation observed at 1 mM MgCl_2_ (Figure 4). Regarding the response to classical phosphatase inhibitors, we evaluated the effects of sodium orthovanadate (Van), sodium fluoride (NaF), and ammonium molybdate (Mol) on both Cu^2+^-dependent and Cu^2+^-independent ectophosphatase activities. All these compounds were able to inhibit both activities; however, Cu^2+^-dependent ectophosphatase activity was less sensitive to inhibition (Figure 5).

## 4. Discussion

*Candida albicans* is an opportunistic pathogen capable of causing a wide range of superficial and systemic infections, particularly under immunosuppressive conditions, such as those observed in HIV-positive patients [1,2,3]. In the present study, we identified a marked stimulation of ectophosphatase activity in the presence of micromolar concentrations of CuCl_2_ (Figure 1). Copper modulation has been previously described in phosphatases of other organisms. Beyond the previously mentioned study with *C. parapsilosis* [10], stimulatory effects of copper have been observed on the acid phosphatases of the protists *Trypanosoma brucei* [20] and *Euglena gracilis* [21], the yeast *Yarrowia lipolytica* [22], and the green alga *Pseudokirchneriella subcapitata* [23]. In *Ophiocephalus punctatus* (a freshwater fish), in vivo exposure to copper resulted in the activation of renal alkaline phosphatase [24].

Previous analyses of *C. albicans* isolates from the oral cavities of HIV-positive (HIV^+^) and HIV-negative (HIV^−^) children revealed a strong association between ectophosphatase activity and HIV infection [6]. Interestingly, several studies have reported increased serum copper levels in HIV^+^ individuals, with differences ranging from approximately 1 to 10 µM [25,26,27,28]. Based on our findings, we hypothesize that the elevated ectophosphatase activity observed in HIV^+^ patients may be related to increased copper availability, potentially leading to enhanced enzymatic stimulation. Considering that serum copper levels in healthy individuals range from approximately 20 to 40 µM [25,26,27,28], the increase observed in HIV^+^ individuals is relatively subtle and may not directly justify a stimulation of *C. albicans* ectophosphatase activity. Nonetheless, it is important to emphasize that the periplasmic space constitutes a microenvironment rich in negatively charged components, which may favor the in situ accumulation of Cu^2+^ and thereby contribute to the increase in ectophosphatase activity. However, further studies will be necessary to better understand this possible association.

Regarding the nature of copper modulation, we found that pre-incubating intact cells with reducing agents—except ascorbate—completely prevented the stimulation of ectophosphatase activity induced by CuCl_2_ (Figure 2A). Therefore, this stimulation could be associated with conformational changes resulting from the oxidation of specific amino acid residues, due to the pro-oxidative nature of Cu^2+^ ions. In this scenario, the stimulatory effect would be expected to persist even after the removal of the metal. However, the addition of chelators to cells pre-incubated with the metal (Figure 2B) led to a significant reduction in ectophosphatase activity stimulation, suggesting that this effect is reversible. Furthermore, the pre-incubation of the cells with EDTA yielded similar results to those obtained with the reducing agents, preventing the stimulation of ectophosphatase activity by CuCl_2_ (Figure 2C). It is worth noting that reducing agents such as GSH and DTT can also act as metal chelators. The high nuclear charge of the Cu^2+^ ion favors the formation of a cupric–thiolate complex. GSH, for example, can coordinate and form a Cu^2+^-GSH complex in different conformations [29,30,31]. Considering that copper stimulation is a reversible effect and that only the reducing agents containing sulfhydryl groups were able to prevent the stimulation, we suggest that the action of Cu^2+^ on *C. albicans* ectophosphatase does not appear to involve a redox mechanism. However, since only ascorbate was tested as a non-thiol reducing agent, further studies using additional compounds are needed to fully confirm this hypothesis.

Since transition metals can induce oxidative stress through the production of hydroxyl radicals via the Fenton reaction [15,32], the stimulation of ectophosphatase activity could be associated with the presence of reactive oxygen species (ROS). If this hypothesis were correct, the generation of ROS due to the simultaneous presence of ascorbate and Cu^2+^ in the medium would be expected to increase the stimulation induced by the metal, as previously reported [33]. However, the level of stimulation promoted by the metal remained unchanged regardless of the presence or absence of ascorbate, allowing us to rule out this possibility (Figure 2A).

Therefore, the stimulation of ectophosphatase activity is likely dependent on the interaction between ectophosphatases and the Cu^2+^ ion. Considering that the primary ligands of Cu^2+^ ions in proteins are the imidazole ring of histidine residues and the thiol group of cysteine or methionine residues [34], it is plausible that such interactions induce conformational changes in ectophosphatases that enhance catalysis.

We observed significant differences between the biochemical properties of basal (Cu^2+^-independent) and Cu^2+^-dependent ectophosphatase activities. Since the experiments were conducted using intact cells, two possible explanations arise: (1) both activities are associated with the same enzyme, or (2) there are at least two distinct ectophosphatases in *C. albicans*. Considering the first hypothesis, the basal activity may reflect enzyme forms located in microenvironments less accessible to the metal ion or associated with cell wall components that limit their interaction with copper. However, given the distinct biochemical profiles observed, we consider it more plausible that two different enzymes are involved. Further studies using molecular biology and genetic manipulation techniques will be necessary to fully elucidate this question.

Cu^2+^-dependent ectophosphatase activity has a slightly alkaline optimum pH (7.6), while Cu^2+^-independent activity is much more effective at acidic pH (Figure 3). The existence of two enzymatic forms with different pH optima suggests a functional adaptation of *C. albicans* to different host niches, where pH can vary significantly. Cu^2+^-dependent activity also exhibited a significant and dose-dependent stimulation in the presence of MgCl_2_, while Cu^2+^-independent ectophosphatase activity was insensitive to this metal (Figure 4). This finding suggests a two-level regulatory mechanism, in which copper acts initially by promoting a conformational or functional change in the enzyme, then enabling the modulating action of magnesium. The modulation promoted by synergistic interaction between metal ions has been previously reported in phosphatases and other enzymatic activities [35,36].

Although both basal and Cu^2+^-independent and Cu^2+^-dependent ectophosphatase activities were inhibited by vanadate, sodium fluoride, and molybdate, Cu^2+^-dependent ectophosphatase activity presented lower sensitivity to all the inhibitors tested (Figure 5). This may reflect a conformational modulation promoted by copper binding, altering the active site microenvironment or the accessibility of inhibitors. In other enzyme systems, changes in cofactors have been attributed to changes in the main by-products or substrates [35]. The lower sensitivity observed may represent a functional advantage, allowing the enzyme to maintain its catalytic activity even in the presence of inhibitory compounds in the host extracellular environment.

In several studies [6,8,9,10,11,12], the irreversible profile of enzyme inhibition produced by vanadate has enabled the comparison of the ability of fungal cells to attach to host cells when ectophosphatase activity was fully functional versus when surface enzyme activity was inhibited by pretreatment with vanadate. The results suggest that ectophosphatase activity contributes to the adhesion process, potentially playing a key role in establishing infection [6,8,9,10,11,12]. In *C. albicans*, the pretreatment of yeast cells with vanadate significantly reduced their attachment to epithelial cells [6].

One of the most efficient host responses to fungal infections is a process termed nutritional immunity, a defense strategy that limits pathogen access to essential micronutrients or exposes them to toxic metal concentrations [37]. A widely recognized concept in innate immunity is host-imposed copper toxicity, in which elevated copper concentrations act as an effective biocide against microbial pathogens [38]. In this context, *C. albicans* also expresses cytosolic metallothioneins and a P1-type ATPase responsible for extruding Cu^2+^ ions across the plasma membrane, allowing fungal cells to survive even when exposed to high copper concentrations [16]. On the other hand, although serum copper levels increase, copper can become limiting in the kidney, the major site of *C. albicans* infection [39]. To overcome this condition, *C. albicans* readily adapts by increasing copper uptake and by exchanging metal cofactors for antioxidant SODs [2]. At the membrane level, copper exhibits high-affinity binding to phosphatidylserine residues, which can result in membrane damage. Accordingly, the plasma membrane architecture of *C. albicans* appears to be structured to protect the cell surface from copper-induced attack [14].

Taken together, our findings demonstrate that copper stimulates ectophosphatase activity in *C. albicans*, adding to our understanding of how this enzyme may function under copper fluctuations. Since previous studies have reported that ectophosphatases are involved in host–pathogen interactions, the copper-induced stimulation observed here could contribute to these processes. Further studies are needed to clarify this potential role.

## Figures and Tables

**Figure 1 pathogens-14-00667-f001:**
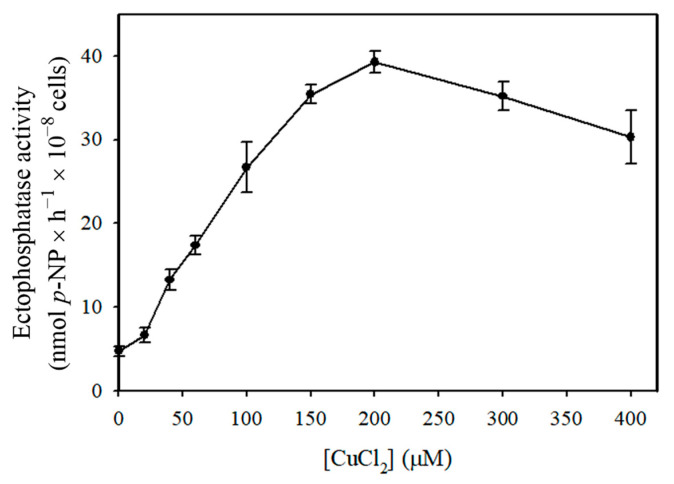
Influence of increasing CuCl_2_ concentrations on ectophosphatase activity of *Candida albicans.* Ectophosphatase activity was determined in the presence of increasing CuCl_2_ concentrations (0–400 µM). Values are expressed as the mean ± standard error from at least three independent experiments performed in triplicate.

**Figure 2 pathogens-14-00667-f002:**
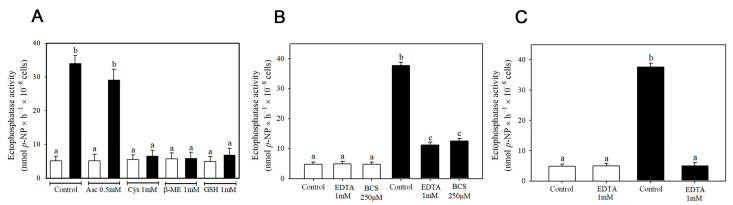
Effect of reducing agents and metal chelators on the CuCl_2_-induced stimulation in ectophosphatase activity of *Candida albicans.* Fungal cells were pre-incubated with the reducing agents—ascorbate (Asc), cysteine (Cys), reduced glutathione (GSH), and β-mercaptoethanol (β-ME)—for 30 min prior to the start of the reaction. In the control group, no reducing agents were added. Ectophosphatase activity was measured in the absence (white bars) or presence of 200 µM CuCl_2_ (black bars), which was added 15 min after the addition of reducing agents (**A**). *C. albicans* cells were pre-incubated with buffer in the absence (white bars) or presence of 200 µM CuCl_2_ (black bars) for 30 min prior to the start of the reaction. After 15 min of pre-incubation, the chelating agents EDTA (1 mM) or BCS (250 µM) were added. Following an additional 15 min, ectophosphatase activity was measured in the absence (controls) or presence of the respective chelating agents (**B**). Fungal cells were pre-incubated with EDTA (1 mM) for 30 min prior to the start of the reaction. Control conditions were pre-incubated in the absence of the chelator. Ectophosphatase activity was then measured in the absence (white bars) or presence of 200 µM CuCl_2_ (black bars), which was added 15 min after the addition of EDTA (**C**). Values are expressed as the mean ± standard error from at least three independent experiments performed in triplicate. Different lowercase letters indicate statistically significant differences (*p* < 0.05).

**Figure 3 pathogens-14-00667-f003:**
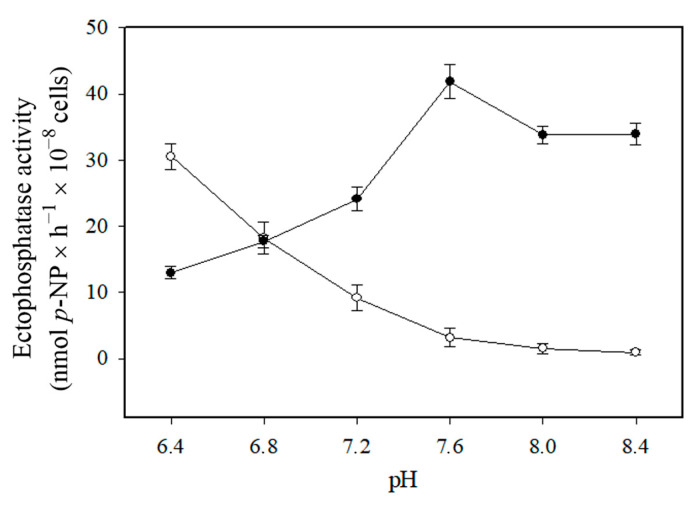
Effect of pH on Cu^2+^-independent and Cu^2+^-dependent ectophosphatase activities of *Candida albicans.* Ectophosphatase activity was measured at different pH values in the absence or presence of 200 µM CuCl_2_. Cu^2+^-dependent ectophosphatase activity (black circles) was calculated by subtracting the Cu^2+^-independent activity (white circles) from the activity measured in the presence of 200 µM CuCl_2_. Values are expressed as the mean ± standard error from at least three independent experiments performed in triplicate.

**Figure 4 pathogens-14-00667-f004:**
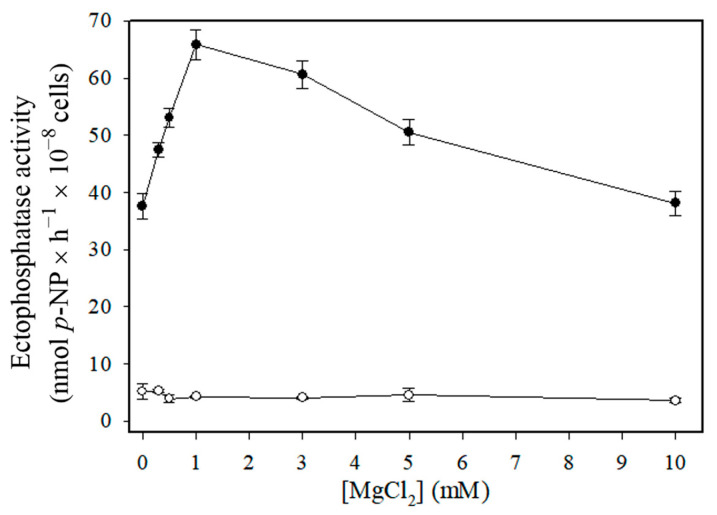
Influence of increasing concentrations of MgCl_2_ on Cu^2+^-independent and Cu^2+^-dependent ectophosphatase activities of *Candida albicans.* Ectophosphatase activity was measured in the presence of increasing MgCl_2_ concentrations (1–10 mM), in the absence or presence of 200 µM CuCl_2_. Cu^2+^-dependent ectophosphatase activity (black circles) was calculated by subtracting the Cu^2+^-independent activity (white circles) from the activity measured in the presence of 200 µM CuCl_2_. Values are expressed as the mean ± standard error from at least three independent experiments performed in triplicate.

**Figure 5 pathogens-14-00667-f005:**
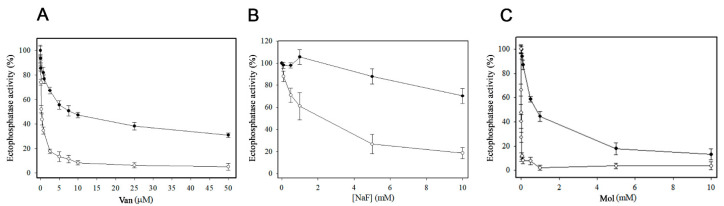
Influence of classical phosphatase inhibitors on Cu^2+^-independent and Cu^2+^-dependent ectophosphatase activities of *Candida albicans.* Ectophosphatase activity was measured in the presence of increasing concentrations of classical phosphatase inhibitors—sodium orthovanadate (Van; 1–50 µM, (**A**)), sodium fluoride (NaF; 1–10 mM, (**B**)), and ammonium molybdate (Mol; 1–10 mM, (**C**))—in the absence or presence of 200 µM CuCl_2_. Cu^2+^-dependent ectophosphatase activity (black circles) was calculated by subtracting the Cu^2+^-independent activity (white circles) from the activity measured in the presence of 200 µM CuCl_2_. Values are expressed as the mean ± standard error from at least three independent experiments performed in triplicate.

## Data Availability

The data that support the findings of this study are available in Section 2.

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
