# Peer review of "Copper-Induced Stimulation of Ectophosphatase Activity of Candida albicans"

_pathogens, 2025, doi:10.3390/pathogens14070667_

Round 1
Reviewer 1 Report
Comments and Suggestions for Authors
The MS by Freitas-Mesquita and co-workers describes how the activity of C. albicans ectophosphatases are modulated by copper. The MS is well written and approaches a very relevant thematic, although hardly explored, highlighting the importance of pathogen’s/fungal’s ectophosphatases not only in the biochemistry of C.albicans but also in the mechanisms by which these fungi can strive the host immune system. The results clearly indicate the regulation of ectophosphatase by copper and the authors very well discuss this biological trait in terms of the pathogenesis of C. albicans, including the sites of infection. For sure this will open novel perspectives to understand different issues that make of C. albicans a successful colonizer and an opportunistic agent of local o systemic infection.
Minor comments:
Fig. 2 – indicate the meaning of a, b and c (maybe in the Materials and Methods section…)
Line 283 . rephrase to “… kidney, the major target of disseminated C. albicans infection [REF; e.g. Jawale & Biswas, 2021]
Author Response
#R1: Fig. 2 – indicate the meaning of a, b and c (maybe in the Materials and Methods section…)
Response: The lowercase letters (a, b, c) indicate statistically significant differences (p < 0.05). This information was only available in the figure legend. As suggested, we added this information in the Materials and Methods section of the revised manuscript (lines 108-109).
#R1: Line 283. rephrase to “… kidney, the major target of disseminated C. albicans infection [REF; e.g. Jawale & Biswas, 2021]
Response: As suggested, we added this reference to the sentence (line 309).
Reviewer 2 Report
Comments and Suggestions for Authors
Overall, the research work is conducted adequately. However, I suggest you complete some experiments or limit your conclusions. I'll explain later.
Line 33: Define WHO
Line 56-60: Complete this idea.
Lines 71-72: Be more specific about these growth factors.
Line 75: Perform the necessary calculations to change from rpm to x g.
In the introductory section or in the section on the microorganisms used, please provide a little more detail about the strain used for the experiments and the reasons for that genetic background. Would the results be the same if you had used the wild type? Why not conduct the experiments with both (WT and auxotrophic) and see the results?
Lines 213-216. Please take these lines with more caution, and emphasize that further experiments are needed to corroborate this suggestion.
Lines 230-232: Take these lines with caution, since only ascorbate was used; some other compounds that do not have sulfhydryl groups could have been used.
Lines 268-271: References of these studies.
Line 291: The experiments conducted do not support the conclusion that the enzyme studied is involved in adhesion. Other types of experiments are needed to confirm this.
Lines 210-216: Mention the Cu levels under these conditions.
In some parts of the text, when referring to the enzyme, it seems to refer to two different enzymes, and in others, to the same enzyme but with two functions. I suggest you restructure these ideas so that it is understood that it is the same enzyme and that it can perform a second function at different pH levels and concentrations of cofactors.
Author Response
Reviewer 2
#R2: Line 33: Define WHO
Response: As suggested, we defined WHO as World Health Organization (line 33).
#R2: Line 56-60: Complete this idea.
Response: As suggested, we have expanded the paragraph by including examples of Candida albicans regulatory mechanisms involved in maintaining copper homeostasis (lines 60-63).
#R2: Lines 71-72: Be more specific about these growth factors.
Response: In the original manuscript, we did not mention the uridine supplementation. Therefore, we have rewritten the paragraph to correct this, adding the proper information (lines 78-79).
#R2: Line 75: Perform the necessary calculations to change from rpm to x g.
Response: As suggested, we have revised the centrifugation parameter to specify 1500 × g instead of rpm (line 81).
#R2: In the introductory section or in the section on the microorganisms used, please provide a little more detail about the strain used for the experiments and the reasons for that genetic background. Would the results be the same if you had used the wild type? Why not conduct the experiments with both (WT and auxotrophic) and see the results?
Response: The CAI-4 strain of Candida albicans was used in this study because it is a well-characterized laboratory strain derived from the clinical isolate SC5314, retaining key pathogenic properties in experimental models. Its genetic construction (URA3-) facilitates molecular manipulations, which is an important property for enabling future studies aimed at clarifying the relevance of copper stimulation in ectophosphatase activity and its potential involvement in fungal infection. Although we recognize that experiments using the wild-type strain could also provide valuable insights, CAI-4 was chosen here due to its experimental tractability. As suggested, we have added a sentence in the Materials and Methods section to clarify the reason for choosing this strain in the present study (lines 75-77).
#R2: Lines 213-216. Please take these lines with more caution, and emphasize that further experiments are needed to corroborate this suggestion.
Response: As suggested, we have rewritten this paragraph in the revised manuscript to highlight further studies are necessary to better understand this possible association (lines 217-230).
#R2: Lines 230-232: Take these lines with caution, since only ascorbate was used; some other compounds that do not have sulfhydryl groups could have been used.
Response: The conclusion described is supported by the results shown in Figures 2A and 2C. The fact that the stimulation is reversible is a critical observation, indicating that copper’s effect is due to its presence and binding rather than an irreversible oxidative (redox) modification of the target. To justify why some reducing agents were also able to prevent this stimulation, we hypothesize that they act as chelators due to their sulfhydryl groups, which explains why ascorbate does not present the same effect. However, we understand that further tests are necessary to fully validate this hypothesis. Therefore, we have rewritten the end of the paragraph to address the reviewer’s comments (lines 244-249).
#R2: Lines 268-271: References of these studies.
Response: These references were available at the end of the next sentence. As suggested, we also included them right after the expression "in several studies" (line 293).
#R2: Line 291: The experiments conducted do not support the conclusion that the enzyme studied is involved in adhesion. Other types of experiments are needed to confirm this.
Response: Although our experiments did not directly assess fungal adhesion, previous studies have reported that ectophosphatases are involved in host-pathogen interactions, suggesting that the copper-induced stimulation observed here could contribute to these processes. As suggested, we have rewritten this paragraph in the revised manuscript to clarify the actual contribution of our data to this field (lines 314-319).
#R2: Lines 210-216: Mention the Cu levels under these conditions.
Response: As suggested, we mentioned the copper levels under the conditions described and discussed the possible impacts of this increase on the modulation of ectophosphatase activity (lines 220-229).
#R2: In some parts of the text, when referring to the enzyme, it seems to refer to two different enzymes, and in others, to the same enzyme but with two functions. I suggest you restructure these ideas so that it is understood that it is the same enzyme and that it can perform a second function at different pH levels and concentrations of cofactors.
Response: Since our experiments were conducted using intact cells, two possible explanations arise: (1) both activities are associated with the same enzyme, or (2) there are at least two distinct ectophosphatases in C. albicans. Considering the first hypothesis, the basal activity may reflect enzyme forms located in microenvironments less accessible to the metal ion or associated with cell wall components that limit their interaction with copper. However, given the distinct biochemical profiles observed, we consider it more plausible that two different enzymes are involved. Further studies using molecular biology and genetic manipulation techniques will be necessary to fully elucidate this question. Following the reviewer’s comment, we have included this discussion in the manuscript (lines 263-272) and also complemented the final paragraph of the Introduction to indicate that the existence of two different enzymes represents the most plausible hypothesis (lines 69-70).
Round 2
Reviewer 2 Report
Comments and Suggestions for Authors
Thank you very much for responding to the corrections and taking the suggestions into account for your work.